# Applying the principles for digital development to improve maternal and child health in the Peri-urban areas of Karachi, Pakistan

**Hareem Ahmer**[1], **Kinza Farooqui**[1], **Karim Jivani**[1], **Rehan Adamjee**[2], **Zahra Hoodbhoy**[3]*

**1** Data and Digital, Vital Pakistan Trust, Karachi, Sindh, Pakistan, **2** MBA and Public Policy Graduate Student, Harvard Business School & Harvard Kennedy School, Boston, Massachusetts, United States of America, **3** Department of Pediatrics and Child Health, Aga Khan University, Karachi, Sindh, Pakistan

* zahra.hoodbhoy@aku.edu

## Abstract

Low- middle-income countries, including Pakistan, are facing significant obstacles in their efforts to achieve the global targets for maternal, newborn, and child health (MNCH) defined by the Sustainable Development Goals. Barriers at the individual, health system, and contextual levels undermine healthcare access for pregnant women and children, disproportionately affecting those in low-resource settings. To address these challenges in the high-mortality, peri-urban areas of Karachi, VITAL Pakistan Trust and Aga Khan University launched a digital health intervention (DHI) to stimulate demand for health services and streamline care management for health workers at the primary care level. In this case study, we present a narrative review of the design, development, and deployment of the DHI, an Android-based application, in accordance with the Principles for Digital Development. We draw on the initial experience with implementation to reflect on how each of the nine Principles was considered during different phases of the project lifecycle, focusing on the lessons learned and challenges encountered during this process. By engaging with end-users and understanding the community, we were able to map existing relationships and workflows onto a digital platform to address major challenges hindering service delivery. Leveraging insights from field observations and user feedback, we collaborated with experts in healthcare and technology to develop the DHI, which has now scaled to 44 peri-urban settlements in Karachi. Our experience underscores the value of substantiated frameworks like the Principles. However, on-ground challenges reveal important caveats requiring further assessment. These include building community trust in new digital systems and ensuring the ethical use of health data, particularly in low digital and data literacy contexts. Based on this understanding, we share recommendations for conditions central to the effective integration and uptake of technology in healthcare, specifically within the context of digital health for MNCH.

**Data Availability Statement:** All data are in the manuscript.

**Funding:** Funding support for this work was from the Bill and Melinda Gates Foundation (BMGF) with

grant number INV-057220 (USD 3,700,000 to ZH). The funder had no role in study design, data collection and analysis, decision to publish, or preparation of the manuscript.

**Competing interests:** The authors have declared that no competing interests exist.

## Author summary

Unlocking the potential of digital health requires careful consideration of the way digital tools are developed and deployed. The Principles for Digital Development provide nine evidence-based guidelines for integrating technology into development practice. When applied to the digital health sector, they can help improve the usability, acceptability, and impact of technology-enabled care. We applied the Principles within the scope of maternal, newborn, and child health (MNCH), launching a DHI for health workers in high-mortality, peri-urban settlements in Karachi. This case study highlights our experience designing, developing, and deploying the DHI to address major barriers in healthcare within these settlements. We provide insights and recommendations on navigating the use of technology in the delivery of MNCH services. Through our findings, we aim to strengthen the evidence base for the implementation of DHIs in low-resource settings, thus building a foundation for the broader MNCH community, especially within the Pakistani context where digital health is still in its infancy. Other initiatives, both within and outside of MNCH, can use these findings to maximize their impact and further evaluate the implications of DHIs in transforming health outcomes.

## Introduction

Low- middle-income countries (LMICs) including Pakistan are lagging in achieving the maternal, newborn, and child health (MNCH) targets set by the Sustainable Development Goals [1]. Limited access to healthcare facilities, fragmented healthcare services, and shortages of human and material resources compromise the coverage and quality of care for pregnant women and children, increasing their risk of adverse health outcomes [2]. These challenges can be addressed through digital health interventions (DHIs) that stimulate demand for health services through behavior change communication; streamline service delivery by facilitating task management and tracking of care providers; and provide health workers with swift access to health information for clinical decision-making, even in remote settings [3,4]. Recognizing this potential, the World Health Assembly has endorsed the development and use of innovative technologies to advance "good health and well-being for all" [5].

There is a growing body of work on DHIs in all fields of medicine, including MNCH, which has assessed the scope, feasibility, and impact of existing interventions in LMICs. Colaci et al. identified areas such as health education, data collection, provider communication, appointment reminders, and clinical follow-ups that have benefitted from technology [6]. Kabongo et al. reported that the use of DHIs has the potential to enhance healthcare satisfaction and support for pregnant women and mothers, thus promoting the utilization of antenatal and postnatal care services [7]. However, although numerous projects have been piloted over the past several years, few have managed to successfully integrate with existing health systems. DHI implementation remains a complex and challenging process, particularly in hard-to-reach communities struggling with poor infrastructure and technological capacity [7].

Unlocking the potential of DHIs calls for careful consideration of how they are developed and deployed within existing health systems. Many challenges associated with the use of new technologies can be addressed, if not avoided altogether, through thoughtful design that meets the needs of users and health systems [8]. Frameworks guiding the planning process can help public health researchers and practitioners deliver technology-enabled care that is accessible, convenient, and effective [9]. The Principles for Digital Development are an example of a

**Table 1. Principles for Digital Development.**

|   | Principles for Digital Development |
|---|---|
| 1 | Design with the user |
| 2 | Understand the existing ecosystem |
| 3 | Design for scale |
| 4 | Build for sustainability |
| 5 | Be data-driven |
| 6 | Use open standards, open data, open source, and open innovation |
| 7 | Reuse and improve |
| 8 | Address privacy and security |
| 9 | Be collaborative |

framework that offers evidence-based guidelines for developing and implementing DHIs in practice [10].

The Principles for Digital Development were established in 2014 by the *Principles for Digital Development Working Group*—a multilateral group of donors, implementers, and stakeholders from organizations such as UNICEF, UNDP, WHO, USAID, and the Bill and Melinda Gates Foundation [10]. They offer nine recommendations for technology-enabled interventions, providing a guiding framework for development practitioners working in resource-constrained environments to increase the likelihood of program success (Table 1). Over the years, the Principles have been applied to initiatives across several programmatic areas, including climate change [11], education [12], gender equality [13], and youth social entrepreneurship [14]. Within the scope of healthcare, practitioners have used the Principles to inform mobile health interventions aimed at improving disease reporting and detection [15], hospital and health center quality [16], and maternal and neonatal health systems [17], among other objectives.

This article presents a case study of a DHI that allows healthcare workers to track and deliver an integrated continuum of MNCH services in the peri-urban communities of Karachi, Pakistan [18]. We reflect on our experience designing, developing, and deploying the DHI in accordance with the Principles for Digital Development to better understand how technology can be leveraged to support safe, accessible, and coordinated healthcare for pregnant women and children. Lessons learned from this experience have the potential to inform intervention development and implementation planning in other low-resource settings [19], driving the impact of digital health practice and policy.

## Presentation of the case

We draw on data and observations generated through practice to formalize our learnings during the design, development, and initial deployment of the DHI.

### Implementation context

VITAL Pakistan Trust (VPT) is a non-profit organization created to support MNCH research and service delivery in the high-mortality areas of Karachi, Pakistan. In 2014, VPT partnered with the Department of Pediatrics and Child Health at Aga Khan University (AKU) to implement a MNCH program in Rehri Goth, a peri-urban slum located in Bin Qasim town, Karachi. The total population of the area is currently 88,359, with 18,966 married women of reproductive age (MWRA) and 13,624 children under five years old. Access to the nearest public

hospital, located 25 kilometers away, is severely constrained due to various social, economic, and infrastructural barriers that limit women's mobility. As a result, women in this community tend to rely on informal healthcare services offered by untrained and unregulated providers during pregnancy and delivery.

To address this issue, the MNCH program established a primary health center (PHC) with the aim of enhancing access to formal healthcare services within the community. The PHC involves several teams working across the continuum of care for MNCH, including antenatal, postnatal, and family planning services for married women of reproductive age and immunization and physician services for children under the age of five. Despite initial success, including a near 50% drop in neonatal mortality, reductions in health indicators plateaued by 2018, calling for the incorporation of new tools and innovation within existing processes to improve health and well-being in the community.

## DHI description

The DHI was launched in 2020 in partnership with VentureDive, a leading software company in Pakistan [20]. It is an Android-based application designed to support care coordination by consolidating longitudinal patient data and streamlining communication across maternal and newborn care teams, ultimately facilitating the provision of high-quality MNCH services both at the community and facility levels. Currently, the application is divided into 14 integrated components, referred to as "modules," that are organized to align with the specific tasks and workflows involved at each stage of the pregnancy, postpartum, and newborn care journey (Fig 1).

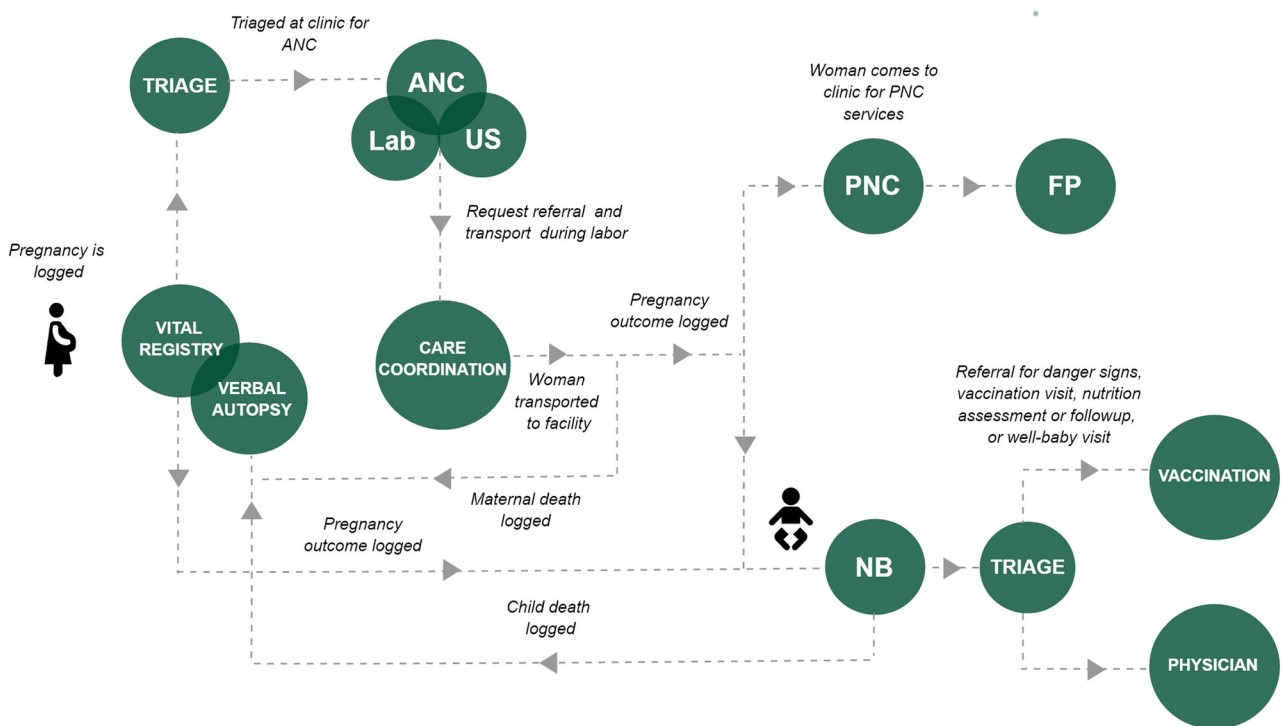

**Fig 1. DHI workflow in the community.** The following abbreviations represent the system modules covering different phases of the MNCH care journey: ANC = Antenatal Care, Lab = Laboratory Investigations, US = Ultrasound, PNC = Postnatal Care, FP = Family Planning, NB = Newborn Care.

The user base includes community health workers (CHWs), midwives, allied health workers (phlebotomists and sonologists), vaccinators, and physicians. All end-users possess basic adult literacy skills. CHWs are hired from within the community and have at least 10 years of schooling. Midwives have diplomas with at least 2 years of training, whereas physicians are required to have at least a bachelor's degree. Due to religious and cultural sensitivities, patients prefer seeking care from female healthcare providers; as such, the healthcare workforce is composed entirely of female professionals.

The application allows health workers to record and retrieve key clinical data across multiple touchpoints within a single system, facilitating the tracking of a woman's and child's complete medical journey and improving the delivery of care. Each touchpoint in the workflow represents a module and associated end-user with a distinct role in the continuum of care for MNCH (Fig 1). For example, the antenatal care components shown in Fig 1 depict five end-users engaged in a linear, task-based workflow. Patients visiting the PHC begin their care journey with the triage user, who captures their clinical measurements and forwards their antenatal assessment task to the midwife. The midwife user performs the assessment and creates ultrasound and lab tasks for the sonologist and phlebotomist, respectively. If the patient requires management from a higher care facility, the midwife user can also generate a task for the care coordinator, who manages referrals to secondary and tertiary healthcare facilities.

The DHI's workflow can be modified, and modules included or excluded based on project requirements and operational objectives. This became pertinent when the DHI was scaled to other communities where clinical operations were limited. For instance, vaccination services are offered in isolation in certain super high-risk communities, so only the vaccinator module of the application has been implemented for use. New variables can also be added to existing modules to support the needs of clinical research. One such example is PRISMA (Pregnancy Risk, Infant Surveillance, and Measurement Alliance), an ongoing cohort study being conducted in Rehri Goth and Ibrahim Hyderi that aims to evaluate pregnancy risk factors using longitudinal patient data [21]. To support study requirements, additional questions have been added to the midwife, ultrasound, lab, and newborn modules, specifically for the end-users at these sites.

The DHI was built on the Open Smart Register Platform (OpenSRP), an open-source mobile health platform designed to support the WHO SMART guidelines for data-driven decision-making in healthcare [22]. Key features of the DHI include searchable task lists for easy task management; logs for outcomes of interest, requests for ultrasounds and lab investigations, and referrals to support cross-team communication; longitudinal woman and child profiles to ensure data reliability and robustness; and high-risk flags for pregnancy and malnourished children to guide health workers (Table 2; Fig 2).

Data entered into the DHI is ingested into an open-source NoSQL document database. To allow for quick monitoring and reporting, data is further extracted, loaded, and transformed into a PostgreSQL relational database management system through which dashboards and scheduled reports are created and shared with relevant stakeholders.

## DHI impact

The DHI has helped streamline and strengthen services for the provision of basic and comprehensive maternal and newborn care. Targeted interventions based on insights from the DHI have enabled end-users to expand their outreach within the community. For instance, outreach efforts for family planning are largely informed by the data collected through the DHI. Outreach teams utilize the DHI to compile lists of women with 3 or more children under the age of five. This helps identify households where there may be an unmet need for family

Table 2. DHI features.

| System Goal | System Feature |
|---|---|
| Task management support to organize, track, manage, and automate work. | • Ordered and searchable system-generated task lists for routine and repeatable workflows.<br>• Flags to indicate the urgency of task(s). |
| Cross-team communication support through logs, requests, referrals | • Standard protocols to notify relevant teams of high-priority events such as logs for new births.<br>• Easy referrals to higher care facilities via outgoing requests to care coordination midwives.<br>• Ability to view the status of pending requests. |
| Consistent profiles across time and care modules | • Creation and maintenance of longitudinal profiles for women and children.<br>• Data sharing across modules to reduce data conflicts and data collection time.<br>• Roles and permissions allow checks before data can be overwritten. |
| Clinical decision support | • Risk scores for pregnant women and malnourished children allow flagging and management of high-risk cases.<br>• All care teams have access to easily searchable, customized profiles at critical decision junctures, i.e., antenatal, intrapartum, and postnatal care visits. |
| Easy retrieval of patient records | • Multi-variable search makes it easy to find a woman or child.<br>• Key data is accessible to health workers or midwives, reducing dependency on data recall by pregnant women. |
| Replicate desirable offline elements | • Profile and screen views that provide useful snapshots.<br>• Permissions and roles allow control over data.<br>• Aggregate data is regularly synced and easily viewable. |
| Provide functionality for migration and adoption | • Easy support for location updates and migration of families.<br>• Unlink feature for mothers and children in case of adoption. |

planning. Similarly, CHWs, who conduct daily household visits as part of routine monitoring and surveillance, use the DHI to reinforce the importance of formal healthcare in the community by registering and referring pregnant women and children to the PHC.

While formal research on the impact of the DHI on end-users already exists [23], further research assessing its impact on health-seeking behavior and maternal and newborn outcomes is to be conducted in due course. Below, we detail our experience applying each of the nine Principles during different phases of the product lifecycle, focusing on the lessons learned and challenges encountered during this process.

## Intervention elements

### 1. Design with the user

Understanding user perspectives regarding their current scope of work and its challenges is an essential step while re-engineering processes [10]. As a first step, our program and development teams engaged with different cadres of the field staff through field observations and face-to-face interviews to understand the nature of their work (which tasks were time-consuming, which processes could be moved online versus offline), the tools used to conduct these tasks, and their technological readiness.

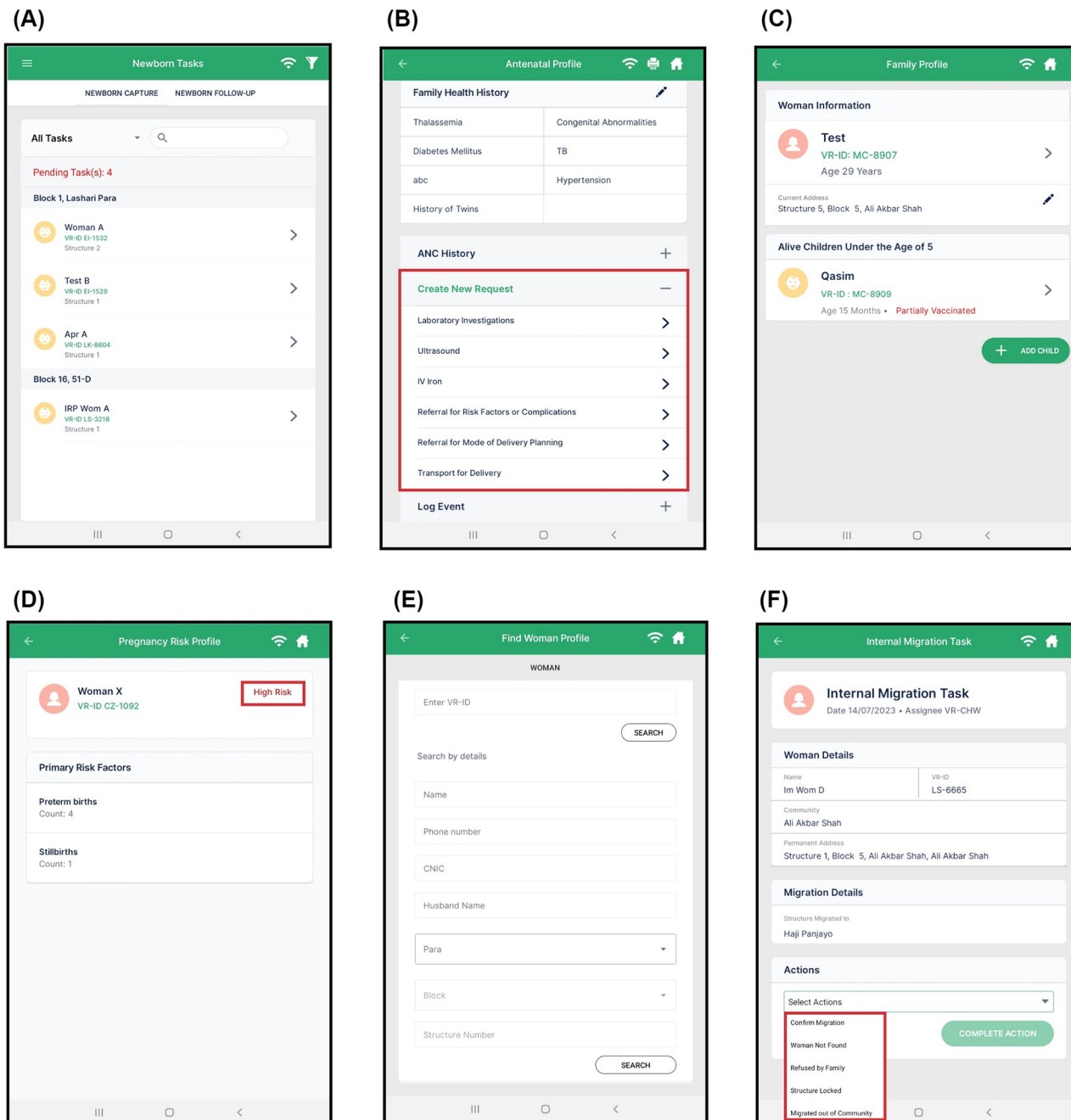

**Fig 2. DHI features and screenshots.** (A) Task list for newborn capture, (B) Request ultrasound, lab investigation, or referral (C) Longitudinal mother and child profiles, (D) Risk flags for pregnancy, (E) Multi-variable search, (F) Migration updates.

This needs assessment revealed that field staff were utilizing paper registers and worksheets to store patient information, which was both time-consuming and prone to errors. Informal methods such as WhatsApp messages and phone calls were used to relay key information between home-based and facility-based teams. Although most members of the field staff were active users of social media and mainstream apps, few had experience with digital health tools. Based on these findings, our development team created design mockups for each module and

revisited the field to gather further feedback. This process provided insight into how potential design options (with respect to structure, functions, style, and content) would affect current clinical workflows before progressing to the development phase. Building on this understanding, our development team devised a detailed plan for the development of an application that replicated desirable elements of paper-based systems while improving the continuity of care through real-time and near-real-time data access (Table 2; Fig 2).

Designing with the user is an incremental and iterative process (10), calling for the routine evaluation of user needs and preferences to ensure that DHIs are making a meaningful impact. To understand the user experience, a mixed methods study was conducted six months after implementation. Findings revealed that although health workers were satisfied in terms of content quality, aesthetics, and ease of use, poor internet connectivity and issues with data synchronization were compromising service quality [23]. The team has since been actively engaged in streamlining technological support to improve user performance and experience with the new system. This involved establishing dedicated IT and product teams responsible for implementing a digital feedback system to create effective feedback loops. Site-specific WhatsApp groups, including both end-users and product experts, were used to implement this feedback system. Direct communication ensured the regular flow of feedback regarding bugs, issues, and operational challenges, allowing our team to transform problem areas on the ground into actionable insights and improve the user journey. An essential part of this process was closing the feedback loop by informing users of solutions and changes implemented based on their prior reports, fostering trust in the system and the development process.

## 2. Understand the existing ecosystem

An understanding of the local context and systems within which health services are delivered is essential to develop an effective digital solution [10]. The Department of Pediatrics and Child Health at AKU has been actively involved in improving MNCH services in peri-urban low-income settlements along the coastal region of Karachi since 2003 [24,25]. In these communities, pregnant women and newborns are monitored through household surveillance conducted by trained community health workers. If necessary, they are referred to PHCs or hospitals for further treatment. The PHCs are run by AKU and staffed with midwives, physicians, vaccinators, and other healthcare providers. This long-standing presence in the community has allowed the department to gain deep experiential knowledge about the realities of working in the field and implementing healthcare programs in resource-constrained environments. At the primary level, low- and mid-level care providers struggled with high volumes of tasks, incomplete or inaccurate medical records, and a heavy reliance on human recall and judgment for decision-making. Relevant clinical assessments and information within existing MNCH workflows were needed, including flags for high-risk pregnancies based on predefined factors, so that health workers could provide personalized evidence-based care.

Moreover, although internet coverage has increased in Pakistan over the years, the availability of high-performance internet remains unequal across the rural and urban divide, with poor speed and quality in the former [26]. Due to unstable internet services in the community, we chose a hybrid online-offline setup so that users could enter data offline, which would be automatically synced once connected to a network server.

## 3. Use open standards, open data, open source, and open innovation

In response to the structural limitations identified above, our development team used OpenSRP's system architecture, which supports hybrid operations and enables performance management and streamlined data sharing [22]. To ensure effective data management, we

used PostgreSQL, leveraging its advanced data types and performance optimization features for data integrity and consistency. Adopting open-source software helps reduce licensing and support costs, which is an important consideration when implementing DHIs in low-resource settings. Improving the affordability of technical infrastructure has helped create the necessary foundation for the future scalability of the DHI.

## 4. Reuse and improve

Flexibility and adaptability are core facets of the DHI to keep up with user feedback and varying research project requirements where variables can be modified based on study objectives. Moreover, our development team adapted aspects of the OpenSRP systems to match contextual needs. An example includes the childhood immunization domain, which is a singular module within the OpenSRP system architecture [27]. However, the current DHI divides this module into "mobilizer" and "vaccinator" to support VPT's strategy for vaccination outreach.

Performance upgrades such as a real-time synchronization feature were added to the DHI's basic features to ensure efficient information flows between users. This feature minimized the time taken for tasks, requests, and logs to reach the relevant team. Similarly, patient profiles were contextualized to user roles, which reduced data storage and memory needs and ensured that only relevant information was visible to each user.

## 5. Be data-driven

A data-driven approach to development means leveraging data to inform decision-making at all levels throughout the project lifecycle [10]. The core purpose of this DHI was streamlining data collection, storage, and use to promote high-quality MNCH service delivery. The DHI content includes evidence-based items from the WHO Manual for Positive Pregnancy Experience [28] and Integrated Management of Neonatal and Childhood Illnesses (IMNCI) guidelines [29] to ensure standardized service delivery. The availability of real-time patient data helps midwives quickly and accurately identify high-risk pregnancies, newborn care teams track and manage children with poor growth trajectories, and vaccinators target unimmunized cases. Interactive data visuals created from this dataset further assist project teams in making data-informed decisions. For example, the immunization team utilizes a web-based dashboard to track efforts at the district, town, and site level. Visualizing vaccination progress allows supervisors to identify gaps and refine outreach strategies to target missed settlements and areas with poor coverage, thus strengthening accountability, responsiveness, and performance over time.

## 6. Address privacy and security

The DHI's security works on three different layers: network, data, and device. For the former, the system supports 'Secure Socket Layer' (SSL) protocols for data encryption across internet servers. Secondly, to maintain data security, database access is always authenticated with only authorized users being allowed to view and manipulate data. Lastly, the DHI utilizes SQLCipher, an open-source database extension, which encrypts local database files on devices and thus reduces vulnerability to malicious applications and data forensic tools in cases of device theft or loss.

Privacy protocols are also upheld to ensure the protection of both user and patient information during routine data sharing with partners. For instance, the DHI generates unique alphanumeric IDs against all registered women and children to prevent duplications and anonymize patient data to maintain confidentiality during any data sharing with research

partners. The DHI has also embedded questions to enroll eligible participants into research studies once written and verbal informed consent is obtained.

## 7. Design for scale

In a short span of 30 months, this DHI has been scaled to 44 peri-urban areas across Karachi, Pakistan, serving over 150,000 women and children. Building a DHI where flexibility for adaptation was embedded into the design was key to an eventual scale-up strategy. With minimal impact on the DHI's overall functionality, modules can be added or removed based on changes in context and clinical workflows. For example, the vaccinator modules are available in isolation in areas where immunization services are being offered independently.

## 8. Build for sustainability

Initiatives are more likely to have long-term, positive impacts if they are set in policies, daily practices, and user workflow [30]. As discussed under Principles 1 and 2, our program team recognized the importance of conducting user research to align user needs with technological possibilities. Sustainability also requires quantifying long-term costs and deciding investments in terms of people, money, technology, and institutional capacity [10,30]. Based on early discussions on the long-term costs of the project, the team opted for locally available and less costly but reliable hardware (tablets and smartphones) and open-source software for both data collection and management, thus promoting sustainability with lower maintenance costs.

The DHI is currently being supported by a variety of internationally funded projects. However, it is worth acknowledging that mobilizing predictable sources of financing remains a challenge in the field of global health [31]. To improve financial security in the future, the DHI could possibly be monetized by charging research partners a data management fee, while the larger good of the community in mind.

## 9. Be collaborative

This DHI is an apt example of leveraging input from diverse experts across healthcare and technology, i.e., VPT, AKU, and VentureDive, to ensure evidence-based implementation of the DHI. The DHI also remains central to key local and national collaborations, playing a pivotal role in service delivery. For example, VPT's immunization program employs a synergetic model with the Expanded Program of Immunization (EPI) and Polio Eradication Initiative (PEI) to strengthen routine vaccination coverage for unvaccinated children. Similarly, the Family Planning (FP) program has been implemented in collaboration with the Population Welfare Department of the Sindh Government.

## Case discussion

Using the Principles for Digital Development has helped ensure initial success in our work. By engaging with the end-user and understanding the community, we were able to map team relationships and information flows to relevant technology and address major barriers hindering service delivery. When planning for future DHIs, it would be beneficial for implementers to conduct a systematic investigation into why digital solutions have not been adopted in the first place. Chattopadhyay devised a framework to survey a set of Indian rural primary health centers and explore available resources in terms of the perceptions of healthcare staff and ICT support at the organizational level [32]. Conducting such surveys should be the first step toward identifying and addressing infrastructural challenges, including lack of access to the internet, requisite software, and a constant source of electricity [32]. This approach is reflected

in "understanding the ecosystem," which is one of the first Principles [10]. The initial design and development of our DHI also incorporated a similar strategy, where we worked to ensure the feasibility of adopting digital interventions within the community.

Previous studies have highlighted the advantages of a contextually driven design process for the implementation of DHIs. Haroun et al. emphasize the importance of involving local communities and health system actors to co-create digital interventions with good acceptability [33]. Drawing on early research for Afya-Tek, a DHI designed to enhance referral tracking and care coordination for maternal, child, and adolescent health in Tanzania, they argue that it allows implementers to gain a comprehensive understanding of challenges associated with both the use and non-use of digital health tools [33]. Similarly, Chamberlain et al. attribute the success of Ananya, a mobile health communication program designed to improve RMNCH practices in the north Indian state of Bihar, to a user-centered design approach [34]. This approach helped align digital services with community needs, building a strong foundation for scalability, sustainability, and health system integration [34].

The effective use of DHI has been associated with enhanced performance among health workers, leading to improved outcomes in MNCH in other LMICs [35]. One notable example is Mobile for Mothers (MfM), an app deployed in rural India to assist CHWs in registering pregnancies and births, providing counseling to pregnant women regarding antenatal, intra-partum, and postnatal care, as well as collecting data during each home visit to optimize services [36]. This initiative resulted in an increase in maternal health knowledge, thereby improving women's likelihood of attending a minimum of four antenatal care visits and delivering at a health facility [36,37]. Within our context, the vast amount of high-quality, multidimensional, and longitudinal MNCH data gathered through the DHI can be leveraged further to develop and validate pregnancy risk stratification models based on composite risk factors. Such advancements have significant implications for service delivery in LMICs, where a high burden of maternal and newborn mortality is compounded by inadequate resources and a shortage of trained health workers [38]. The ability to predict a pregnant woman's risk of adverse health outcomes through contextually relevant decision support can ultimately facilitate resource efficiency in patient management, ensuring that each woman receives the right level of care at the right time, thus improving outcomes [38,39].

However, in evaluating our experience, it is also crucial to consider the challenges that affect the integration and uptake of DHIs in LMICs. By utilizing these findings, we can gain valuable insights into how the Principles can be better prioritized to advance technology-enabled care for marginalized and vulnerable populations. For example, despite collaborating with health workers to create a platform tailored to their needs and capabilities, gaining their trust has been challenging in communities with low digital literacy and digital self-efficacy. When faced with network failure or application issues, health workers tend to revert to paper-based registers due to an inability to independently troubleshoot and fix software or hardware-related problems. Analyses of Edtech interventions in LMICs have emphasized the role of continuous training and responsive support resources in encouraging end-users to adopt new technologies in their work [40]. Similar steps, such as developing a troubleshooting guide and training end-users to address common problems [41], should be taken to make health workers more digitally literate, and therefore confident, within the context of their work.

The increasing digitalization of health data in environments with low digital literacy also raises concerns about consent, choice, privacy, ownership, and accountability [42]. Encouraging the ethical use of digital data to improve health outcomes requires the establishment of strong data governance structures, which involve defining processes, roles, policies, and standards for data collection, storage, access, and sharing [43]. Achieving consensus on these definitions, however, requires alignment between different stakeholders with different experiences

and perspectives on data. This is difficult in LMICs where significant digital inequality leads to disparities in knowledge that undermine the ability of more vulnerable populations, as data subjects, to participate in decision-making about data privacy and security [42,43]. Concomitant efforts to improve data literacy for healthcare and advocate for data governance regulations at the individual, organizational, and government levels can facilitate the ethical and appropriate use of health data in the long run.

Financial sustainability is also a concern in the global health community where many programs rely on grant-based funding to support improvements in healthcare while lacking strategies to sustain them once the funding is over [31]. Given insufficient government health spending [31], the achievement of long-term gains in health–particularly through DHIs– requires sustainable business models that create alternative revenue streams and gradually reduce dependence on donors [44]. This transition should, however, be accompanied by a rigorous analysis of the effect of monetizing social interventions on value creation for marginalized and vulnerable populations, emphasizing transparency, local ownership, equity, and safety as key considerations [45].

It is important to acknowledge that DHIs are not all-round solutions to structural and systemic barriers in healthcare; rather, they are better classified as part of the solution. Treating technology as the answer to complex human problems is an oversimplified approach to DHIs as technology has a social dimension, which can potentially resist or reproduce inequalities [46]. Integrating situated critique, contextualization, and human-centered design can help reduce technological solutionism and make DHIs more impactful to their intended audience [46].

## Conclusion

Adopting evidence-based practices is essential for successfully implementing DHIs into the existing health ecosystem. In our context, the Principles provided a guiding lens to outline core contextual barriers and integrate relevant solutions. However, as observed in the planning and implementation of this DHI, even with the use of substantiated frameworks such as the Principles, healthcare delivery in low-resource settings comes with unique challenges and considerations This study provides evidence for leveraging digital technology to transform maternal and child health outcomes, particularly in the context of Pakistan, where the digital health ecosystem is still in its infancy.

## Acknowledgments

We would like to acknowledge the Venture Dive, Aga Khan University, and VITAL Pakistan Trust teams in the development and deployment of the digital health intervention described in this manuscript.

## Author Contributions

**Conceptualization:** Hareem Ahmer, Kinza Farooqui, Rehan Adamjee, Zahra Hoodbhoy.

**Data curation:** Hareem Ahmer, Kinza Farooqui, Rehan Adamjee.

**Formal analysis:** Hareem Ahmer, Kinza Farooqui.

**Methodology:** Hareem Ahmer, Kinza Farooqui.

**Supervision:** Karim Jivani, Zahra Hoodbhoy.

**Validation:** Karim Jivani, Rehan Adamjee, Zahra Hoodbhoy.

**Visualization:** Hareem Ahmer.

**Writing – original draft:** Hareem Ahmer, Kinza Farooqui.

**Writing – review & editing:** Hareem Ahmer, Kinza Farooqui, Zahra Hoodbhoy.

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
