## [Decision Letter · Decision Letter 0]

29 Aug 2023

PDIG-D-23-00291

Applying the Principles for Digital Development to Improve Maternal and Child Health in the Peri-Urban Areas of Karachi, Pakistan

PLOS Digital Health

Dear Dr. Hoodbhoy,

Thank you for submitting your manuscript to PLOS Digital Health. After careful consideration, we feel that it has merit but does not fully meet PLOS Digital Health's publication criteria as it currently stands. Therefore, we invite you to submit a revised version of the manuscript that addresses the points raised during the review process.

Please submit your revised manuscript within 60 days Oct 28 2023 11:59PM. If you will need more time than this to complete your revisions, please reply to this message or contact the journal office at digitalhealth@plos.org. Please include the following items when submitting your revised manuscript:

We look forward to receiving your revised manuscript.

Kind regards,

Haleh Ayatollahi

Section Editor

PLOS Digital Health

Journal Requirements:

Additional Editor Comments (if provided):

Reviewers' comments:

Reviewer's Responses to Questions

**Comments to the Author**

1. Does this manuscript meet PLOS Digital Health’s publication criteria? Is the manuscript technically sound, and do the data support the conclusions? The manuscript must describe methodologically and ethically rigorous research with conclusions that are appropriately drawn based on the data presented.

Reviewer #1: Yes

Reviewer #2: Partly

Reviewer #3: Partly

2. Has the statistical analysis been performed appropriately and rigorously?

Reviewer #1: N/A

Reviewer #2: N/A

Reviewer #3: N/A

3. Have the authors made all data underlying the findings in their manuscript fully available (please refer to the Data Availability Statement at the start of the manuscript PDF file)?

Reviewer #1: Yes

Reviewer #2: No

Reviewer #3: No

4. Is the manuscript presented in an intelligible fashion and written in standard English?

Reviewer #1: Yes

Reviewer #2: Yes

Reviewer #3: Yes

5. Review Comments to the Author

Reviewer #1: This is an excellent descriptive study of the implementation of electronic tools to improve maternal and child health outcomes. The study and implementation were done based on prior analysis of existing tools and needs. The manuscript is informative, well outlined and clear. 

The figures are almost unreadable, they need to be redone or resubmitted. Figure 4 appears to be the lifecycle of the work. It will benefit from arrows to indicate the progression and its own feedback.

Reviewer #2: Addressing MNCH in the LMI group of nations is important to bridge up the healthcare divide. Therefore, the study is appropriate and pertinent to the present LMI nations' healthcare delivery. Although the study is well-structured, there are the following opportunities to improve it further,

1. Incorporation of Qualitative and Quantitative analysis with reference to end users (demographic and SDoH parameters, comorbidities, addictions, practice of safe sex, etc.)

2. A separate section on how the proposed DHI method has helped generate awareness in the society and how it is measured

3. Authors may be benefited by reading the following articles, which they may cite in this work to strengthen its foundation:

Chattopadhyay S., Daneshgar F. – “An Awareness Net Collaborative Model for Schizophrenia Management”, International Journal of Advanced Intelligence Paradigms (2013); 5(3): 217-232. 

Acharya U. R., Faust O., Ghista D. N., Sree V. S., Alvin A.P.C., Chattopadhyay S., Lim T-C., Ng E. Y. K., Yu W.Y – “A Systems Approach to Cardiac Health Diagnosis", International Journal of Medical Imaging and Health Informatics (2013); 3: 1-7. 

Chattopadhyay S., – “A Prototype Depression Screening Tool for Rural Healthcare: A Step towards e-Health Informatics”, Journal of Medical Imaging and Health Informatics (2012); 2(3): 244-249 

Chattopadhyay S., Saurabh R., Land L., Acharya U. R. – “Studying Infant Mortality Rate: A Data Mining Approach”, Health and Technology (2011); 1(1): 25-34 

Daneshgar F., Chattopadhyay S. –“A Framework for Crisis Management in Developing Countries”. Intelligent Decision Technologies: an international journal (2011); 5(2): pp. 189-199. 

Chattopadhyay S. –"A Framework for Studying Perceptions of Rural Healthcare Staff and Basic ICT Support for e-health Use: An Indian Experience". Telemedicine and e-Health (2010); 16(1): pp. 80-88. 

Li J, Land L.P.W, Ray P. Chattopadhyay S. –"E-Health Readiness Framework from Electronic Health Records Perspective". International Journal of Internet and Enterprise Management: Special Issue in Healthcare (2010); 6(4): pp. 326-348

Ray P., Chattopadhyay S. – “Fuzzy Awareness Model for Disaster Situations”. Intelligent Decision Technologies: an international journal [Special Issue on Intelligent Decision Making in Dynamic Environments: Methods, Architectures and Applications] (2009); 3 (1): pp. 75-82.

Reviewer #3: Thank you kindly for the opportunity to read through the paper entitled, “Applying the principles for digital development to improve maternal and child health in the peri-urban areas of Karachi Pakistan.” This paper is well-written and easy to follow and describes important work into using novel digital technologies to improve the health of women and children. I have made a few comments regarding how this paper might be strengthened below. 

First, I would suggest that the Principles for Digital Development section in the Materials and Methods be moved to the Introduction section. I think an upfront discussion of these principles and why they are appropriate to guide the development of the DHI presented in this paper is important. 

I would also recommend adding more details to describe the tool. In particular, I think it should be abundantly clear when reading through the Materials and Methods what comprises this tool and how its functionalities might improve maternal and Child health. For instance, I looked to figure 1 and noted discussion of a triage component of healthcare providers as well as a care coordination component but it was not clear to me how these pieces of healthcare workflow are supported (if at all) by this DHI. I was also not clear as to what the 14 integrated modules involve. More detail might also be added regarding how this DHI can be customized to project requirements and objectives. Can detail be provided on which projects the team might be describing here? 

Throughout the Results section, I felt much more detail could be added. 

- For instance, in Section One Design with User, I was not clear as to how user centered design was conducted in this particular instance. Who was involved in the process? How many people were involved in the process? When did the process occur? Where did the process occur? And how were users opinions solicited? 

- I also felt that this section did not highlight to any great degree how codesign principles were embedded in this project. I know some early consultation with users occurred, but how were users [if at all] continually engaged in the process over time? 

- The sentence “the team has since been actively engaged in streamlining technological support to improve performance and experience with the new system” is vague to me. Details related to how this project was carried out are needed to improve the chances that it can support others aiming to engage in similar work. 

- As another example, In Section Two Understand the Existing Ecosystem, the authors describe some findings related to local challenges, but it is not clear to me how these challenges were identified and prioritized by eventual DHI end users. Similar to the above, I am curious to know how these challenges were identified? Who was involved in the process? Etc. 

Do the authors have any comments they can make on how well these processes worked? As the study was focused on implementation, I would like more details on who is using this product, how well it is being used, how care outcomes are changed, etc. 

Overall, I think the authors have undertaken important work here. To maximize the potential impact this work can have on others interested in learning about or following the path of this team, many more details about the DHI in question, how this work was done, and the impact that it has had are needed.

6. PLOS authors have the option to publish the peer review history of their article (what does this mean?). If published, this will include your full peer review and any attached files.

**Do you want your identity to be public for this peer review?** For information about this choice, including consent withdrawal, please see our Privacy Policy.

Reviewer #1: Yes: Laritza M. Rodriguez

Reviewer #2: Yes: Subhagata Chattopadhyay

Reviewer #3: No

---

## [Decision Letter · Decision Letter 1]

9 Oct 2023

PDIG-D-23-00291R1

Applying the Principles for Digital Development to Improve Maternal and Child Health in the Peri-Urban Areas of Karachi, Pakistan

PLOS Digital Health

Dear Dr. Hoodbhoy,

Thank you for submitting your manuscript to PLOS Digital Health. After careful consideration, we feel that it has merit but does not fully meet PLOS Digital Health's publication criteria as it currently stands. Therefore, we invite you to submit a revised version of the manuscript that addresses the points raised during the review process.

Please submit your revised manuscript within 60 days Dec 08 2023 11:59PM. If you will need more time than this to complete your revisions, please reply to this message or contact the journal office at digitalhealth@plos.org. Please include the following items when submitting your revised manuscript:

We look forward to receiving your revised manuscript.

Kind regards,

Haleh Ayatollahi

Section Editor

PLOS Digital Health

Journal Requirements:

Additional Editor Comments (if provided):

Although the topic of the manuscript is interesting, there are some major issues which need to be addressed:

As noted by the authors “a narrative review of the design, development, and deployment of the DHI” has been presented in this manuscript. Therefore, the methods and results sections do not show any details for a research study and these headings should not be included in the manuscript. Overall, it is better to present it as a case study, and address all methodological principles for it.

Reviewers' comments:

Reviewer's Responses to Questions

**Comments to the Author**

1. If the authors have adequately addressed your comments raised in a previous round of review and you feel that this manuscript is now acceptable for publication, you may indicate that here to bypass the “Comments to the Author” section, enter your conflict of interest statement in the “Confidential to Editor” section, and submit your "Accept" recommendation.

Reviewer #1: (No Response)

Reviewer #2: All comments have been addressed

Reviewer #3: All comments have been addressed

2. Does this manuscript meet PLOS Digital Health’s publication criteria? Is the manuscript technically sound, and do the data support the conclusions? The manuscript must describe methodologically and ethically rigorous research with conclusions that are appropriately drawn based on the data presented.

Reviewer #1: Yes

Reviewer #2: Yes

Reviewer #3: Yes

3. Has the statistical analysis been performed appropriately and rigorously?

Reviewer #1: Yes

Reviewer #2: N/A

Reviewer #3: N/A

4. Have the authors made all data underlying the findings in their manuscript fully available (please refer to the Data Availability Statement at the start of the manuscript PDF file)?

Reviewer #1: Yes

Reviewer #2: (No Response)

Reviewer #3: Yes

5. Is the manuscript presented in an intelligible fashion and written in standard English?

Reviewer #1: Yes

Reviewer #2: Yes

Reviewer #3: Yes

6. Review Comments to the Author

Reviewer #1: The figures improved but the caption numbers on the figures are lost!

Please put in more thought on the figures. There is a lot of white space wasted and some of the figures provide very little information. Figure 2 might benefit from changing the layout to portrait with two screenshot views on each row and arrows to indicate the flow. Remember the figures have to stand on their own and be informative.

Reviewer #2: The paper looks better and accepted for publication. 

Congratulations to the authors!

Reviewer #3: Thank you kindly for the time and attention spent addressing each of my comments.

7. PLOS authors have the option to publish the peer review history of their article (what does this mean?). If published, this will include your full peer review and any attached files.

**Do you want your identity to be public for this peer review?** For information about this choice, including consent withdrawal, please see our Privacy Policy. 

Reviewer #1: Yes: 

Reviewer #2: Yes: Subhagata Chattopadhyay

Reviewer #3: No

---

## [Decision Letter · Decision Letter 2]

19 Dec 2023

Applying the Principles for Digital Development to Improve Maternal and Child Health in the Peri-Urban Areas of Karachi, Pakistan

PDIG-D-23-00291R2

Dear Dr. Hoodbhoy,

We are pleased to inform you that your manuscript 'Applying the Principles for Digital Development to Improve Maternal and Child Health in the Peri-Urban Areas of Karachi, Pakistan' has been provisionally accepted for publication in PLOS Digital Health.

Best regards,

Haleh Ayatollahi

Section Editor

PLOS Digital Health

Reviewer Comments (if any, and for reference):

Reviewer's Responses to Questions

**Comments to the Author**

1. If the authors have adequately addressed your comments raised in a previous round of review and you feel that this manuscript is now acceptable for publication, you may indicate that here to bypass the “Comments to the Author” section, enter your conflict of interest statement in the “Confidential to Editor” section, and submit your "Accept" recommendation.

Reviewer #1: All comments have been addressed

Reviewer #2: All comments have been addressed

Reviewer #3: All comments have been addressed

2. Does this manuscript meet PLOS Digital Health’s publication criteria? Is the manuscript technically sound, and do the data support the conclusions? The manuscript must describe methodologically and ethically rigorous research with conclusions that are appropriately drawn based on the data presented.

Reviewer #1: Yes

Reviewer #2: Yes

Reviewer #3: Yes

3. Has the statistical analysis been performed appropriately and rigorously?

Reviewer #1: Yes

Reviewer #2: Yes

Reviewer #3: N/A

4. Have the authors made all data underlying the findings in their manuscript fully available (please refer to the Data Availability Statement at the start of the manuscript PDF file)?

Reviewer #1: Yes

Reviewer #2: Yes

Reviewer #3: Yes

5. Is the manuscript presented in an intelligible fashion and written in standard English?

Reviewer #1: Yes

Reviewer #2: Yes

Reviewer #3: Yes

6. Review Comments to the Author

Reviewer #1: Thank you for the submission and the added effort in clarifying the figures and tables. This tool promises to be a helpful add-on to the well being and monitoring of pregnant women in peri-urban communities.

Reviewer #2: The article, after revisions, look professional and sound. It is accepted for publication from my side.

Reviewer #3: Thank you for addressing my comments.

7. PLOS authors have the option to publish the peer review history of their article (what does this mean?). If published, this will include your full peer review and any attached files.

**Do you want your identity to be public for this peer review?** For information about this choice, including consent withdrawal, please see our Privacy Policy.

Reviewer #1: **Yes: **Laritza M Rodriguez

Reviewer #2: **Yes: **

Reviewer #3: No
